# Controlled Synthesis of Tellurium Nanowires and Performance Optimization of Thin-Film Transistors via Percolation Network Engineering

**DOI:** 10.3390/nano15141128

**Published:** 2025-07-21

**Authors:** Mose Park, Zhiyi Lyu, Seung Hyun Song, Hoo-Jeong Lee

**Affiliations:** 1Department of Smart Fab. Technology, Sungkyunkwan University, Suwon 16419, Republic of Korea; pms7879@skku.edu; 2Eastern Institute of Technology, Ningbo 315200, China; zylv@eitech.edu.cn; 3Department of Electrical Engineering, Sookmyung Women’s University, Seoul 04310, Republic of Korea; 4School of Advanced Materials Science and Engineering, Sungkyunkwan University, 2066, Seobu-ro, Jangan-gu, Suwon 16419, Republic of Korea

**Keywords:** tellurium nanowires, polyvinylpyrrolidone concentration, thin-film transistors, percolation network, electrical performance

## Abstract

In this study, we propose a method for systematic nanowire length control through the precise control of the polyvinylpyrrolidone (PVP) concentration during the synthesis of tellurium nanowires. Furthermore, we report the changes in the electrical properties of thin-film transistor (TFT) devices with different lengths of synthesized tellurium nanowires used as channels. Through the use of scanning electron microscopy (SEM) and atomic force microscopy (AFM), it was determined that the length of the wires increased in relation to the amount of PVP incorporated, while the diameter remained consistent. The synthesized long wires formed a well-connected percolation network with a junction density of 4.6 junctions/µm^2^, which enabled the fabrication of devices with excellent electrical properties, the highest on/off ratio of 10^3^, and charge mobility of 1.1 cm^2^/V·s. In contrast, wires with comparatively reduced PVP content demonstrated a junction density of 2.1 junctions/µm^2^, exhibiting a lower on/off ratio and reduced charge mobility. These results provide guidance on how the amount of PVP added during wire growth affects the length of the synthesized wires and how it affects the connectivity between the wires when they form a network, which may help optimize the performance of high-performance nanoelectronic devices.

## 1. Introduction

Te nanowires (Te NWs) have attracted a lot of attention as material for next-generation nanoelectronic device due to the excellent electrical [1] and optical properties [2] exhibited by their special geometry. In device research using these nanowires, aspect ratio control directly affects the properties of nanowire devices. Especially in thin-film transistors (TFTs) [3], the on/off ratio, conductivity, and carrier mobility [4], which are the key performance parameters, are greatly affected, so accurate aspect ratio control plays a major role in optimizing nanowire device properties. However, traditional synthesis methods, such as chemical vapor deposition (CVD) [5], physical vapor deposition (PVD) [6], and hydrothermal techniques [7], present challenges in precisely controlling the length and diameter of nanowires [8]. This variability in device properties poses a significant challenge to the fabrication of high-performance semiconductor devices using Te NWs.

In this regard, the role of surfactants such as PVP, which can effectively control the morphology of nanowires [9], has been emphasized and developed in recent research on solution-based nanowire synthesis methods. In studies such as Liu et al.’s [10], it has been demonstrated that PVP plays a role in promoting the growth of Te NWs by controlling the growth direction and morphology of the nanowires [9,10,11,12,13,14,15,16,17,18]. However, it remains challenging to carefully control the length of the nanowires and maintain the same diameter.

Additionally, when adjusting the length of nanowires, it is essential to consider how it will affect the percolation network when integrated into a TFT device. From a percolation perspective, network morphology analysis is important for characterizing electrical devices, particularly TFT devices [19,20]. Percolation networks provide a continuous path for electron transport with overlapping, connected nanowire junctions [21,22]. The efficiency of charge transfer in this percolation network is strongly influenced by the nanowire’s length [23], contact density [24], and spatial distribution [25]. Optimizing these factors can significantly enhance device performance, including improvements in conductivity, on/off ratio, and carrier mobility [26]. Despite extensive research on percolation networks [27], studies specifically focused on Te NW networks are limited. In this regard, a notable gap exists in research regarding how different PVP concentrations (hence the nanowire lengths) affect the formation and optimization of percolation networks within TFTs.

In this study, we examined the effect of PVP density on the length of Te NWs while maintaining a consistent diameter. Additionally, we investigated how wire length influences the morphology of the percolation network, as this affects the electrical properties of Te nanowire-based TFT devices. To accomplish this, we used atomic force microscopy to analyze the morphology of the percolation network and the density of its junctions. This analysis provides a comprehensive understanding of how nanowire length and distribution affect electron transport and the electrical properties of devices. The analysis offers detailed insights into precisely controlling nanowire geometry to fabricate high-performance nanoelectronic devices with optimized percolation networks.

## 2. Materials and Methods

### 2.1. Chemicals and Materials

For this study, all chemicals were sourced from Sigma-Aldrich Korea, Seoul, Republic of Korea and were used without any further purification. The materials comprised tellurium dioxide (TeO_2_, purity ≥ 99%), polyvinylpyrrolidone with a molecular weight of 40,000 g/mol, potassium hydroxide (KOH, purity ≥ 99%), ethylene glycol, and L-ascorbic acid (purity ≥ 99%).

### 2.2. Morphology Control Synthesis of Tellurium Nanowires

Following our previously established method [11], we modified the PVP concentration in Solution A. Sample 1 contained 2.5 mmol of PVP (abbreviated as Te-2.5), Sample 2 contained 1.25 mmol of PVP (Te-1.25), Sample 3 contained 0.25 mmol of PVP (Te-0.25), and Sample 4 contained 0.125 mmol of PVP (Te-0.125). Solution B consisted of 0.3 g of KOH dissolved in 3 mL of ethylene glycol. Solution C was prepared by dissolving 0.2 g of ascorbic acid in 6 mL of deionized water. All other synthesis conditions remained consistent across the four samples. In brief, Solutions A, B, and C were subjected to vacuum degassing for 30 min. Under nitrogen protection, Solution A was gradually added to Solution B, 0.032 mmol of TeO_2_ was added and mixed, and then the mixture was heated to 120 °C. Next, 3 mL of Solution C, preheated to 90 °C, was rapidly introduced into the reaction. The reaction was maintained under nitrogen protection at 120 °C for 24 h.

### 2.3. Device Fabrication

Bottom-gate, top-contact thin-film transistor devices were fabricated on highly doped silicon substrates with a 2000 Å thermally grown SiO_2_ dielectric layer. Gold source and drain electrodes were patterned using thermal electron-beam evaporation through a stainless steel shadow mask, defining a channel length (L) of 1000 µm and a channel width (W) of 50 µm. After electrode formation, 1 µL of an aqueous Te nanowire solution (5 mg/mL) with different morphologies was drop-cast directly between the electrodes. During solvent evaporation, the coffee-ring effect facilitated the self-assembly of the Te nanowires into a percolated network within the channel region.

### 2.4. Material Characterization

X-ray Diffraction (XRD): XRD patterns were collected using a Bruker-D8 DISCOVER instrument (Bruker, Ettlingen, Germany). Field Emission Scanning Electron Microscopy: FESEM images were obtained using a JSM-7600F microscope (JEOL, Tokyo, Japan). X-ray Photoelectron Spectroscopy (XPS): Elemental information was collected using a Thermo ESCALAB 250 HRXPS instrument (Thermo Fisher Scientific, Waltham, MA, USA). Transmission Electron Microscopy (TEM): TEM and HRTEM images were acquired using a JEOL Ltd. JEM 2100F instrument (JEOL, Tokyo, Japan). Optical Bandgap: UV-Vis absorbance spectra were measured in solution using a Cary 5000 UV-Vis-NIR spectrophotometer (Agilent Technologies, Santa Clara, CA, USA). Atomic Force Microscopy: AFM images were collected using a Park Systems AFM instrument to analyze the surface morphology and percolation networks of the tellurium nanowires. Electrical Measurements: Electrical measurements of the fabricated devices were conducted at room temperature in a dark environment using a Keithley 4200B-SCS Parameter Analyzer semiconductor device analyzer (Keithley Instruments, Cleveland, OH, USA).

## 3. Results and Discussion

First, we will discuss the synthesis of Te NWs and the control of their morphology. Precise control of the amount of PVP allows the length of the nanowires to be carefully adjusted while maintaining their diameter. This precise control of nanowire morphology greatly influences the optimization of the electrical properties of nanowire-based thin-film transistors. Figure 1 provides comprehensive information on the material properties of the synthesized Te NWs and confirms successful synthesis. Figure 1a shows the X-ray diffraction (XRD) pattern, which confirms that the Te nanowires were uniformly produced with the desired helical crystal structure and that their crystallinity was preserved during synthesis.

High-resolution transmission electron microscopy (TEM) visualization of the Te nanowire structure (Figure 1c) confirms the well-defined crystal structure, as shown by XRD. The selected area electron diffraction pattern inset also illustrates the nature of the Te lattice. X-ray photoelectron spectroscopy (Figure 1d) confirms the atomic structure of tellurium as well. The characteristic Te peaks at 572.9 and 583.4 eV (Te 3d5/2 and Te 3d3/2, respectively) are present. Additionally, the presence of tellurium oxide is confirmed by the presence of peaks at 576.1 and 586.4 eV. This tellurium oxide is believed to be the result of rapid oxidation in air. A scanning electron microscopy (SEM) image (Figure 1e) reveals the morphology of the synthesized nanowires and demonstrates that nanowires with a uniform diameter, straight orientation, and a large aspect ratio were successfully synthesized. The inverse fast Fourier transform image (Figure 1f) highlights clear lattice fringes, and the lattice spacing distribution (Figure 1g) shows a consistent spacing of 3.3 nm, indicating high crystallinity.

Next, we discussed the influence of PVP concentration on the length and growth of the nanowires. SEM images in Figure 2 show TeNWs synthesized with different PVP concentrations, demonstrating the influence of PVP on nanowire growth. The schematic (Figure 2a) illustrates PVP’s role in regulating nanowire growth, where the PVP (represented by the gray spheres with the hydrophobic tails) surrounds the 1D TeNW (represented by the blue rod), suppressing the growth in the radial direction. The passivation of the TeNW surface leads to a preferential growth in the [001] direction. SEM images of TeNWs synthesized at PVP concentrations of Te-2.5, Te-1.25, Te-0.25, and Te-0.125 are shown in Figure 2b, c, d, and e, respectively, alongside higher magnification images (Figure 2f–i). These images reveal that increasing PVP concentration results in longer nanowires, with average lengths ranging from 4 µm to 14 µm, while maintaining a consistent diameter (Figure 2j).

The optical bandgap and atomic structure properties were analyzed using Tauc plots and XPS, as shown in Figure 3. The Tauc plots (Figure 3a,d) demonstrate that the optical bandgap of Te NWs remained constant despite changes in PVP concentration and nanowire length [28]. This stability of the optical properties is attributed to the control of nanowire morphology (especially the uniformity of diameter) by PVP, which is a crucial consideration for various device applications requiring stable electrical properties. The XRD patterns (Figure 3b,e) once again indicate the invariant of the crystalline phase of the wires as the length varies. In contrast, the XPS spectra (Figure 3c,f) provide details of the binding energies related to the te3d core level, demonstrating the purity and quality of the synthesized nanowires.

After confirming that the variation in PVP concentration can independently control the nanowire length without changing the diameters, which influences the bandgap and crystallinity, we proceeded with AFM analysis to study the formation of the percolation network as a function of nanowire length. The role of nanowire length in forming and optimizing percolation networks within TFTs was investigated using AFM, as shown in Figure 4. We took multiple scans on both the center and edge regions. Representative AFM images of the center (Figure 4a–d) revealed that longer TeNWs (Te-2.5, Te-1.25) formed denser and more interconnected networks, which should enhance the electrical pathways within the device. Around the edges, the “coffee-ring” effect was noticeable as the nanowires are pushed to the boundary of the liquid–surface interface (Figure 4e). The representative AFM images of the edges showed that the thicknesses of the coffee-ring increase as the lengths of TeNWs become shorter (Figure 4f–i). This increase in the coffee-ring effect resulted in uneven distribution (especially noticeable in the shorter TeNW (Te-0.125) samples (Figure 4d,i). This effect indicates that shorter nanowires tend to accumulate at the edges during solvent evaporation, leading to non-uniform film coverage, likely due to their shorter length making them have higher diffusivity in the solvent during the evaporation process.

Figure 5 offers further insight into the surface morphology and nanowire distribution at different regions (center, intermediate, and edge) across samples synthesized with varying PVP concentrations. The AFM topography (Figure 5a) provides a three-dimensional view of the nanowire network, highlighting critical overlap points that contribute to higher junction densities. In order to perform a quantitative analysis, we used custom Python (ver.3.12.2) code to measure junction densities across different regions (center, edge, and intermediate); the result (Figure 5e) indicates that the edge regions, compared to the center, showed much higher junction densities.

As the PVP concentration decreased, the “coffee-ring” effect became more pronounced, and the difference in junction densities between the center and edge regions increased. This effect was most clearly observed in the Te-0.25 sample, where the differences in junction density were most pronounced. However, for the shorter Te-0.125 sample, the nanowires were insufficiently long to establish adequate connections. Consequently, a reduced number of junctions was measured, resulting in reduced electrical properties.

Finally, the electrical properties of the Te nanowire-based TFTs were measured. The electrical characterization of TFTs fabricated using Te NWs was evaluated through key parameters, including the on/off ratio, the drain current, and the carrier mobility (Figure 6). These electrical characteristics were measured ten times for each device. In ordinary cases, when measuring the on/off current and threshold voltage (Vth) to evaluate device performance, the gate voltage range must be regulated to achieve the saturation of the on/off current. However, due to the intrinsic properties of Te nanowires (which can easily absorb and react with humidity in the atmosphere), there was significant hysteresis in the transfer curve, as well as two different threshold points in each direction of the gate voltage sweep. Even after adjusting the gate voltage sweep range, large hysteresis remained, and threshold points appeared near the edge of the sweep range. In this work, we fixed the gate voltage sweep range to start at 50 V and end at −50 V, returning to 50 V. A high saturated current level was considered the on current, and a low saturated current level was considered the off current. Devices fabricated with long nanowires (e.g., Te-2.5) showed improved electrical properties. The on/off ratio was as high as 10^3^, and the charge mobility was 1.1 cm^2^/V·s. Conversely, for short nanowires (Te-0.125), the on/off ratio was less than 10, and mobility was significantly reduced (0.008 cm^2^/V·s). These findings suggest that longer nanowires provide superior percolation pathways, enhanced connectivity, and reduced electrical resistance, which is advantageous for charge transport in TFT devices.

A thorough analysis of junction density, as depicted in Figure 5e, offers insights into the correlation between TFT device properties and junction density. As demonstrated in Figure 6e, the on/off ratio, on-current, and off-current values exhibited a strong dependence on the PVP concentration. This observation indicates that the length of the nanowire plays a significant role in the variation in device properties. The formation of these uniform networks appeared to have contributed to the enhanced electrical properties.

In contrast, the shorter Te NWs, Te-0.25 and Te-0.125, exhibited significantly lower junction densities and more non-uniform junction densities, with a transition in junction density from the center to edge. The Te-0.125 sample demonstrated a low junction density of only 2.1 junctions/µm^2^, which degraded the electrical properties of the device. The findings indicate that the precise length control of the nanowires and the resulting percolation network morphology (i.e., the number of junction densities and uniformities) that can be optimized have a significant impact on the performance of TFT devices.

According to the theory of percolation networks [29], longer nanowires have a reduced percolation threshold concentration and require fewer wires and lower junction density to form a connected network. Furthermore, the additional wires above the percolation threshold concentration form more efficient electron paths and reduce the resistance of the overall network by Kirchhoff’s law. Consequently, the utilization of extended nanowires facilitates enhanced charge transport, enhanced connectivity, and improved electrical properties in comparison to smaller wires with equivalent concentrations. On the other hand, for the shorter-length Te-0.125 sample, the percolation threshold limit is elevated, and even if percolation is achieved, it is difficult to form a well-connected network due to the lack of residual wires, resulting in a decrease in the electrical properties of the device.

## 4. Conclusions

In this study, we successfully synthesized tellurium nanowires through the precise control of PVP concentration while maintaining a consistent diameter. Our results demonstrate that PVP concentration plays a crucial role in determining nanowire length, which, in turn, significantly impacts the formation and optimization of percolation networks within thin-film transistors. The longer Te NWs synthesized with a higher PVP concentration (Te-2.5 sample) formed highly interconnected networks, achieving an average junction density of 4.6 junctions/µm^2^, resulting in superior electrical performance characterized by on/off ratios reaching up to 10^3^ and carrier mobilities of 1.1 cm^2^/V·s. Conversely, the Te-0.125 sample, with shorter nanowires and a reduced junction density of only 2.1 junctions/µm^2^, exhibited significantly lower on/off ratios, below 10. These findings, validated through AFM analysis, highlight the pivotal role of nanowire length and junction density in enhancing the electrical properties of nanoelectronic devices, providing valuable insights for the future design and optimization of high-performance TFTs.

## Figures and Tables

**Figure 1 nanomaterials-15-01128-f001:**
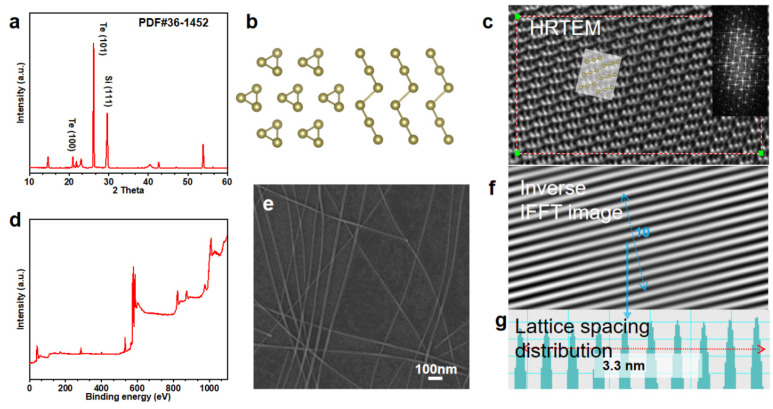
Characterization of synthesized Te nanowires: (**a**) XRD pattern confirming the crystallographic phases of TeNWs, (**b**) visualization of the TeNWs’ crystal structure, (**c**) HRTEM image with an inset of the corresponding SAED pattern, (**d**) XPS spectrum showing elemental composition, (**e**) SEM image illustrating TeNW morphology (scale bar: 100 nm), (**f**) inverse FFT image highlighting lattice fringes, and (**g**) lattice spacing distribution with a measured spacing of 3.3 nm.

**Figure 2 nanomaterials-15-01128-f002:**
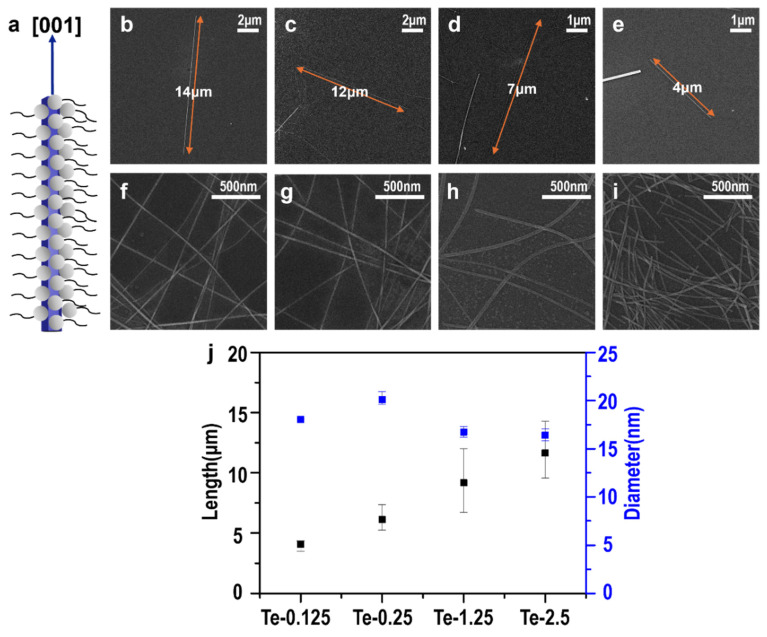
Scanning electron microscopy images of Te nanowires synthesized with different PVP concentrations: (**a**) schematic illustration of PVP’s role in regulating nanowire growth, (**b**,**f**) Te-2.5, (**c**,**g**) Te-1.25, (**d**,**h**) Te-0.25, (**e**,**i**) Te-0.125, and (**j**) length and diameter variations in each synthesized nanowire.

**Figure 3 nanomaterials-15-01128-f003:**
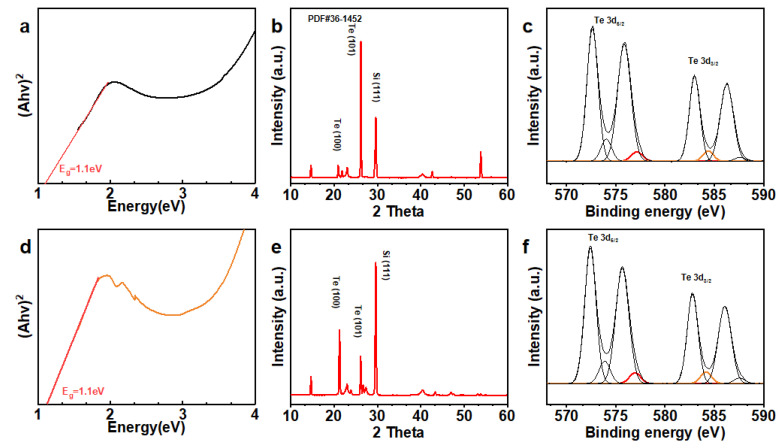
Optical bandgap and structural characterization of Te nanowires synthesized with different PVP concentrations: (**a**) Tauc plot for Te-2.5, (**d**) Tauc plot for Te-0.125, (**b**) XRD pattern for Te-2.5, (**e**) XRD pattern for Te-0.125, (**c**) XPS spectrum for Te-2.5, and (**f**) XPS spectrum for Te-0.125.

**Figure 4 nanomaterials-15-01128-f004:**
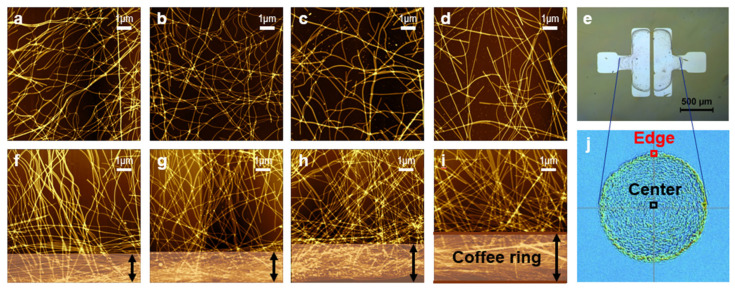
AFM images of Te nanowires deposited with varying PVP concentrations: (**a**,**f**) Te-2.5, (**b**,**g**) Te-1.25, (**c**,**h**) Te-0.25, and (**d**,**i**) Te-0.125 at center (**a**–**d**) and edge (**f**–**i**) regions. Images on the right illustrate the “coffee-ring” effect observed in the Te-0.125 deposition process, with a photograph of the device sample (**e**) and a schematic representation highlighting the center and edge regions (**j**).

**Figure 5 nanomaterials-15-01128-f005:**
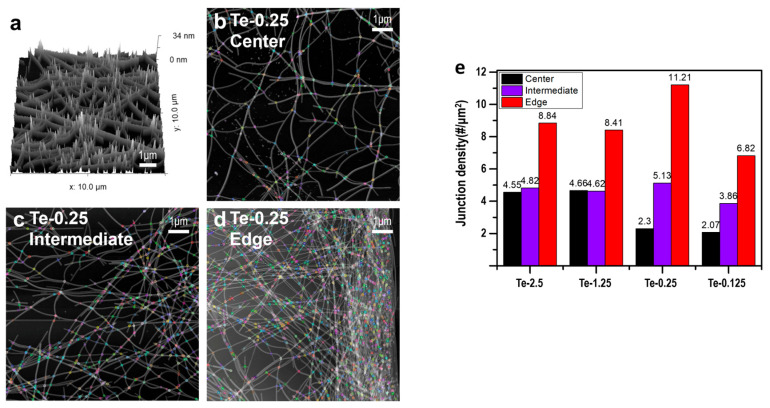
Network morphology and distribution analysis of Te nanowires using AFM and high-resolution imaging: (**a**) 3D AFM topography of the nanowire network, (**b**) nanowire and junction (color dot) distribution at the center region with Te-0.25, (**c**) nanowire and junction distribution at the intermediate region with Te-0.25, (**d**) nanowire and junction distribution at the edge region with Te-0.25, (**e**) bar graph depicting the junction density across different regions (center, intermediate, and edge) for each PVP concentration.

**Figure 6 nanomaterials-15-01128-f006:**
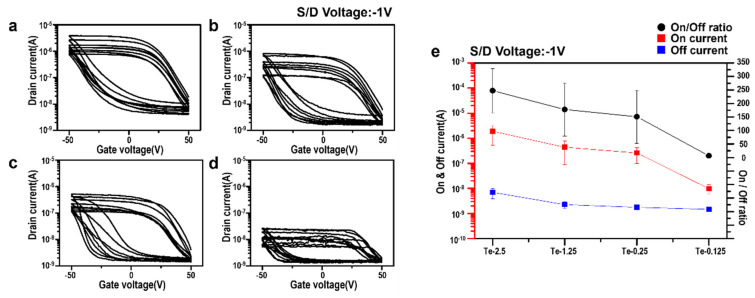
Electrical performance of TFTs with Te nanowires synthesized using different PVP concentrations: (**a**) transfer characteristics (drain current vs. gate voltage) for Te-2.5, (**b**) transfer characteristics for Te-1.25, (**c**) transfer characteristics for Te-0.25, (**d**) transfer characteristics for Te-0.125, and (**e**) the on/off ratio, on current, and off current at a source–drain voltage of −1V for different PVP concentrations (Te-2.5, Te-1.25, Te-0.25, and Te-0.125).

## Data Availability

Data are contained within the article. The data and code that support the findings of this study are openly available in https://github.com/Moses-Park/nanomaterials (accessed on 13 July 2025). The repository contains the full source code to reproduce the results presented in this work. All scripts and related data are licensed under MIT License, allowing for reuse and adaptation with proper attribution.

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
