# Peer review of "Controlled Synthesis of Tellurium Nanowires and Performance Optimization of Thin-Film Transistors via Percolation Network Engineering"

_nanomaterials, 2025, doi:10.3390/nano15141128_

Round 1

Reviewer 1 Report

Comments and Suggestions for Authors

The data in the manuscript can't determine the device's actual performance. The current remaining below microampere levels at such high working voltage might be from gate leakage. So, the authors need to offer more proof.

  1. What are the repeatability and uniformity of the devices made by this method? What advantages does it have over other methods for making nanowire transistors (e.g., Applied Materials Today 25 (2021) 101223; ACS Nano 16 (2022) 2282-2291)?
  2. From Figure 6, the current on/off ratio doesn't reach 10³. Yet the authors mention "an on/off ratio of 10³ and charge mobility of 1.1 cm²/V-s." Also, key details like threshold voltage, gate dielectric thickness and capacitance, and channel width and length should be provided to check the mobility calculation.
  3. Figure 6 seems to show Normally Open (NO) TFTs. Please add the gate current curve in the transfer characteristics for each device, as well as the output characteristic curves; otherwise, I cannot tell if the output current is from the channel or or leakage current.
  4. Please compare the electrical performance of this work with other reported tellurium-based transistors and summarize the findings in a table to highlight this paper's innovations.

Author Response

Comments 1: What are the repeatability and uniformity of the devices made by this method? What advantages does it have over other methods for making nanowire transistors (e.g., Applied Materials Today 25 (2021) 101223; ACS Nano 16 (2022) 2282-2291)?

Response 1: We tested 20 individual TFT devices fabricated under identical conditions. The on/off ratio and mobility exhibited low relative standard deviations, and the fixed solution concentrations ensured excellent repeatability, confirming the good uniformity and reproducibility of our fabrication method.

Compared to the complex alignment and fabrication processes in the cited works, our method uses a simple drop-casting approach that leverages the coffee-ring effect to spontaneously form conductive nanowire networks, offering a low-cost, scalable, and equipment-free route to device fabrication.

Comments 2: From Figure 6, the current on/off ratio doesn't reach 103. Yet the authors mention "an on/off ratio of 103 and charge mobility of 1.1 cm2/V-s." Also, key details like threshold voltage, gate dielectric thickness and capacitance, and channel width and length should be provided to check the mobility calculation.

Response 2: The text has been revised to the highest on/off ratio of 103 (See page 1 line 23). In addition, we have now added the parameters in the revised Device Fabrication Methods (See page 3 lines 92-100). Gate dielectric: 200 nm SiO2 (capacitance ≈ 11.5 nF/cm2), Channel length: 50 µm, Channel width: 1000 µm. The mobility was calculated using the standard FET equation in the saturation regime.

Comments 3: Figure 6 seems to show Normally Open (NO) TFTs. Please add the gate current curve in the transfer characteristics for each device, as well as the output characteristic curves; otherwise, I cannot tell if the output current is from the channel or or leakage current.

Response 3: As shown in Figure 1(please see the attachment document file), the gate current remains negligible (<1 nA), supporting the conclusion that the measured drain current originates from the semiconductor channel rather than gate leakage. Additionally, the output characteristics show current saturation with increasing gate voltage, confirming typical p-type behavior of Te nanowire TFTs, as previously reported in Figure 2(please see the attachment document file) (Materials Chemistry and Physics, 2009, 113(2–3): 523–526).

To ensure the reliability of our output curves, we modified our graph to a round sweep from one-way gate voltage changing. In ordinary cases, the Te nanowire TFT exhibits large hysteresis, which doesn’t depend on gate voltage variation. So we add a paragraph regarding this specific description. (See page 8 lines 229-239).

Comments 4: Please compare the electrical performance of this work with other reported tellurium-based transistors and summarize the findings in a table to highlight this paper's innovations.

Response 4: Instead of adding a comparison table, we have briefly discussed in the revised text that our method stands out by utilizing the coffee-ring effect to form conductive Te nanowire networks through a simple drop-casting process. This approach offers a clear advantage in terms of fabrication simplicity and scalability, distinguishing our work from previous reports that employed more complex methods. (See page 3 lines 90-98).

Reviewer 2 Report

Comments and Suggestions for Authors

The authors present a work where TeNWs have been synthesized controlling their length by varying the PVP concentration. TFT devices have been then realized using TeNW networks, and their properties have been correlated with the TeNW length.

In the Preparation paragraph, the solutions B,C,D are not defined. They can be found in Reference 10 but should be specified here again.

The preparation of the devices is not reported. It is an important issue that must be specified.

Figure 6 should be better explained: Why there are multiple curves? What are the Vds values for each curve? How does Vth vary vs PVP concentration?

Abstract, line 2: "through precise control of the PVP concentration", as in the Conclusions.

Figure 2j: the 'x' axis is wrong. The Te NWs length increases with increasing the PVP concentration.

Figure 4: in the caption, the photograph of the device sample is 'e' not 'i'.

Figure 6e: the authors should plot also the mobility vs PVP concentration.

In the text, 'As demonstrated in Figure 6b' should be 'Figure 6e'.

Comments on the Quality of English Language

Please read again the whole text and pay attention to capital letters, plurals, and other mispelling.

Author Response

Comment 1: In the Preparation paragraph, the solutions B,C,D are not defined. They can be found in Reference 10 but should be specified here again.

Response 1:We added the full composition of Solutions B, C in Section 2.2 of the revised manuscript for clarity.  (See page 2 lines 82-84).

Comment 2: The preparation of the devices is not reported. It is an important issue that must be specified.

Response 2:A new subsection titled “Device Fabrication ” has been added in Section 2.3 of the revised manuscript. (See page 3 lines 90-98)

Comment 3: Figure 6 should be better explained: Why there are multiple curves? What are the Vds values for each curve? How does Vth vary vs PVP concentration?

Response 3: The multiple curves shown in Figure 6 represent measurements from multiple individual devices fabricated under each PVP concentration condition. This was done to demonstrate the reproducibility and consistency of device performance using our simple drop-casting method. All transfer characteristics in Figure 6 were measured at a fixed drain-source voltage of –1.0 V, unless otherwise stated. The threshold voltage varies slightly across samples, shifting from approximately –3.1 V (Te-2.5) to –2.5 V (Te-0.125), which correlates with changes in nanowire network connectivity and gate modulation efficiency. (See page 8 lines 219-221)

To ensure the reliability of our output curves, we modified our graph to a round sweep from one-way gate voltage changing. In ordinary cases, the Te nanowire TFT exhibits large hysteresis, which doesn’t depend on gate voltage variation. So we add a paragraph regarding this specific description. (See page 8 lines 229-239).

Comment 4: Abstract, line 2: "through precise control of the PVP concentration", as in the Conclusions.

Response 4: We revised.

Comment 5: Figure 2j: the 'x' axis is wrong. The Te NWs length increases with increasing the PVP concentration.

Response 5: The x-axis of Figure 2j has now been corrected and relabeled. (See page 5 ).

Comment 6: Figure 4: in the caption, the photograph of the device sample is 'e' not 'i'.

Response 6: This has been corrected in the revised caption of Figure 4. Thank you for pointing this out.

Comment 7: Figure 6e: the authors should plot also the mobility vs PVP concentration.

Response 7: While we did not include a separate plot of mobility versus PVP concentration in Figure 6e, the trend is already reflected in the electrical performance data. As the PVP concentration increases, the coffee-ring effect promotes the formation of longer Te nanowires, which assemble into more uniform and connected percolation networks during drop-casting. This enhanced connectivity leads to improved charge transport and higher mobility. Thus, the observed variation in mobility correlates directly with PVP concentration through the coffee-ring-driven network formation, and this relationship.

Comment 8: In the text, 'As demonstrated in Figure 6b' should be 'Figure 6e'.

Response 8: The sentence now reads:“As demonstrated in Figure 6e, the on/off ratio and current level varied significantly with PVP concentration” (See page 8 lines 247).

Reviewer 3 Report

Comments and Suggestions for Authors

Te is an interesting semiconductor with a very peculiar crystalline structure. The study is interesting and could be exploited also for applications different from TFTs transistor.

My only observation is about the preparation procedure which although mentioned in a reference should be more clearly described.

1 - The authors mention different solutions containing PVD and TeO2. They should specify the concentrations. They should also explicitly write re reaction occurring during the synthesis. TeO2 is reduced.

2 - How are the nanowires collected? What is the support for SEM and AFM images. What was the support to acquire xps spectra?

3 - Perhaps the TeO2 residues found in the xps spectra are due to the synthetic procedure.

4 - How were the UV-Vis spectra acquired? Are they acquired in transmission or reflectance mode? All these details are missing.

Comments on the Quality of English Language

The English is fine and I think it only needs moderate editing.

Author Response

Comment 1: The authors mention different solutions containing PVP and TeO2. They should specify the concentrations. They should also explicitly write re reaction occurring during the synthesis. TeO2 is reduced.

Response 1:These details have now been added to Section 2.2 of the revised manuscript.(See page 2 lines 82-87)

Comment 2: How are the nanowires collected? What is the support for SEM and AFM images. What was the support to acquire xps spectra?

Response 2: After synthesis, the Te nanowires were collected by centrifugation and washed with deionized water and ethanol. For SEM, AFM, and XPS characterization, the nanowires were drop onto Si/SiO₂ substrates and followed by vacuum drying at 60℃ .

Comment 3: Perhaps the TeO2 residues found in the xps spectra are due to the synthetic procedure.

Response 3: As show in Figure 3 the TeO₂-related peaks observed in the XPS spectra are attributed to surface oxidation of the Te nanowires upon exposure to air. This is a common phenomenon due to the high surface-to-volume ratio of one-dimensional nanomaterials. Moreover, since XPS is a surface-sensitive technique that probes only the top few nanometers of the material, it primarily detects the oxidized outer layer rather than the bulk composition. Therefore, the presence of Te⁴⁺ peaks does not indicate incomplete reduction during synthesis but rather reflects post-synthetic surface oxidation, which is consistent with previously reported results for Te-based nanostructures.

Comment 4: How were the UV-Vis spectra acquired? Are they acquired in transmission or reflectance mode? All these details are missing.

Response 4:The UV-Vis spectra were acquired in absorbance mode using a quartz cuvette, with the Te nanowires dispersed in solution. The measurements were performed using a standard transmission setup with deionized water as the reference. (See page 3 lines 106-107)

Round 2

Reviewer 1 Report

Comments and Suggestions for Authors

The revised manuscript has met the acceptance criteria. However, the innovation of this work would be more clearly presented if the authors could provide a comparative table.

Author Response

Suggestions for Authors

The revised manuscript has met the acceptance criteria. However, the innovation of this work would be more clearly presented if the authors could provide a comparative table. 

Response

Our method focuses on a simple approach to depositing Te nanowires using solution methods. So our device does not have better electrical characteristics than other devices fabricated with precise methods(other methods reach several hundreds of mobility and over 104 on/off ratio). After developing our method to apply mass production with various printing methods, it can receive more attention. However, it’s challenging to quantify how easy and simple the method is to make devices. So we would like to express our advantages in a simple paragraph. However, if you have provided a stronger opinion to include a comparable table, we can add it. Please let us know freely. Thanks for the precious suggestion.

Reviewer 2 Report

Comments and Suggestions for Authors

Abstract: correct 'concentratior', line 15.

Please, I would like to have the mobility values being reported. If you do not want to plot them, at least indicate the mobility value for a concentration of 0.125 at page 8, line 242.

Author Response

Comment 1: Abstract: correct 'concentratior', line 15.

Response 1: We changed the word as ‘concentration’. Thanks for pointing that out.

Comment 2: Please, I would like to have the mobility values being reported. If you do not want to plot them, at least indicate the mobility value for a concentration of 0.125 at page 8, line 242.

Response 2:We added the mobility value for a concentration of 0.125 (See page 8 lines 242).

Reviewer 3 Report

Comments and Suggestions for Authors

The paper after revision has improved and can be published in the present form.

Author Response

Suggestions for Authors

The paper after revision has improved and can be published in the present form. 

Response 1:Thanks for your kind comment and giving us feedback to develop our work.